# Analysis of Postural Control in Sitting by Pressure Mapping in Patients with Multiple Sclerosis, Spinal Cord Injury and Friedreich’s Ataxia: A Case Series Study

**DOI:** 10.3390/s20226488

**Published:** 2020-11-13

**Authors:** María Mercedes Reguera-García, Raquel Leirós-Rodríguez, Lorena Álvarez-Barrio, Beatriz Alonso-Cortés Fradejas

**Affiliations:** 1SALBIS Research Group, Nursing and Physiotherapy Department, Faculty of Health Sciences, Universidad de León, 24401 León, Spain; mercedes.reguera@unileon.es (M.M.R.-G.); balof@unileon.es (B.A.-C.F.); 2Faculty of Physical Therapy, Universidade de Vigo, 36156 Pontevedra, Spain; 3Nursing and Physiotherapy Department, Faculty of Health Sciences, Universidad de León, 24401 León, Spain; lalvb@unileon.es

**Keywords:** neurology, pressure ulcer, posture, sitting position, diagnostic equipment, prevention & control, physical therapy specialty

## Abstract

The postural control assessments in patients with neurological diseases lack reliability and sensitivity to small changes in patient functionality. The appearance of pressure mapping has allowed quantitative evaluation of postural control in sitting. This study was carried out to determine the evaluations in pressure mapping and verifying whether they are different between the three sample groups (multiple sclerosis, spinal cord injury and Friedreich’s ataxia), and to determine whether the variables extracted from the pressure mapping analysis are more sensitive than functional tests to evaluate the postural trunk control. A case series study was carried out in a sample of 10 adult patients with multiple sclerosis (*n* = 2), spinal cord injury (*n* = 4) and Friedreich’s ataxia (*n* = 4). The tests applied were: pressure mapping, seated Lateral Reach Test, seated Functional Reach Test, Berg Balance Scale, Posture and Postural Ability Scale, Function in Sitting Test, and Trunk Control Test. The participants with Friedreich’s ataxia showed a tendency to present a higher mean pressure on the seat of subject’s wheelchair compared to other groups. In parallel, users with spinal cord injury showed a tendency to present the highest values of maximum pressure and area of contact. People with different neurological pathologies and similar results in functional tests have very different results in the pressure mapping. Although it is not possible to establish a strong statistical correlation, the relationships between the pressure mapping variables and the functional tests seem to be numerous, especially in the multiple sclerosis group.

## 1. Introduction

Motor control includes all sensory-motor and cognitive processes, through which the neuro-muscular activities employed in a coordinated movement are organized. Thus, it is the integration of all sensory information (both internal and external) to apply the correct combination of muscular solicitation that results in the desired movement. All this requires is the coordinated interaction of the nervous and musculoskeletal systems and is essential for the interaction of the individual with the environment, since, in combination with a voluntary movement action, it is crucial to maintain in control of one’s posture [1,2,3]. Postural control is a complex skill based on the interaction of sensory-motor processes. The main purpose of postural control is to maintain the postural orientation desired by the individual while resisting the force of gravity that acts on their body, as well as to orient the position of the body segments, thus allowing the correct perception of environmental stimuli while performing voluntary tasks [4,5]. Trunk control has been identified as an important early predictor of the functionality of people with neurological disorders and is considered a prerequisite to maintain one’s posture while sitting and standing, to maintain one’s gait while walking, and to simultaneously perform several tasks [6,7].

Most neurological diseases involve the deterioration of motor control, mainly due to the degeneration or injury of the nervous and neuro-muscular systems. These deficits include the loss of coordination and the difficulty in maintaining static posture and gait [5,8]. Movements are often also made less harmonic spatially and/or temporarily by modifications of the neuromuscular system. All this reduces the day-to-day functionality of people with neurological diseases [9,10].

For the correct assessment of postural control, the functional tests and scales available in the clinical setting lack sufficient reliability and sensitivity to small changes in patient functionality [11]. However, in the laboratory environment there are tools capable of quantifying the posture control. These are usually devices of high economic cost and with difficult accessibility in the daily clinic, although they are very reliable and sensitive (i.e., posturography and inertial measurement units) [12,13]. Taking into account this reality, the evaluation of neurological patients with a high degree of dependence is even more problematic, since, even if there are force platforms, electronic gateways, or inertial devices that can quantify the quality of their postural control, these people cannot stand on their own, which is usually the minimum requirement for these devices to perform measurements [14,15].

However, the appearance of pressure mapping has allowed quantitative evaluation of postural control in sitting. This system evaluates the contact surface of the subjects in a wheelchair, thus providing information on their positioning. The use of this tool allows determining which material is more suitable as a seating surface for the individual, providing optimal postural control (within their possibilities) and thus preventing the appearance of pressure ulcers [16,17]. It also allows determining the height and angle of the back of the wheelchair that achieves an optimal pressure distribution [18].

Taking into account all of the above, it was decided to conduct an observational study with neurological wheelchair patients representative of different pathologies that present with postural control disorder (spinal cord injury, multiple sclerosis, and Friedreich’s ataxia) with the following research questions: (a) are pressure mapping results different between patients with different neurological pathologies? and (b) is any variable derived from pressure mapping more sensitive than functional tests usually employed in the clinical setting to evaluate the postural trunk control? In addition, the study works with the initial hypotheses that: (a) the pressure mapping results are different between patients with different neurological pathologies; and (b) at least one variable derived from pressure mapping is more sensitive.

## 2. Materials and Methods

### 2.1. Experimental Design and Sample

A case series study was carried out in a sample of adult patients with neurological involvement and the ability to move from the Center for Disability and Dependency CRE of San Andrés del Rabanedo (Spain). This center has 63 users (institutionalized and outpatient) and all of them were invited to participate in the study. A non-probabilistic sampling of convenience was carried out by selecting the participants who met the following inclusion criteria: (a) having a confirmed medical diagnosis of Multiple Sclerosis, Friedreich’s Ataxia or spinal cord injury that has forced them to use a wheelchair, and (b) being able to maintain the unsupported control of hands in sitting for at least 10 min. Subjects were excluded based on the following criteria: (a) being able to stay in standing position with or without the help of external elements, and (b) presence of cognitive impairment identified by the result of less than 24 points in the Mini-Mental State Examination [19]. After applying these criteria, the sample consisted of 10 subjects (Figure 1).

The participants were previously informed of the objectives of the study. If they agreed to participate, in accordance with the Declaration of Helsinki (rev. 2013), all participants signed an informed consent prior to their participation in the study. The institutional review board approved the study protocol and granted the ethical approval from the Ethics Committee of the University of León (code: ETICA-ULE-035-2018).

### 2.2. Procedure

The participants were contacted and summoned for individual evaluation on a single day and with a five-minute rest interval between each evaluation test. All tests were performed by a physiotherapist previously trained in the evaluation protocol. The evaluation session began with a brief interview in which sociodemographic and anthropometric data (gender, age, weight, height, sitting height, date of neurological diagnosis, and diagnosis of previous and current ulcers) were recorded and the Mini-Mental State Examination was applied. Then, the functional tests detailed below were applied.

Pressure mapping: the Pressure Imaging System X3Display^®^ (XSENSOR, Canada) is a mapping device capable of monitoring the dispersion of pressure between two contact surfaces through the use of sensors [16,18,20]. The system consists of a thin and flexible pillow enclosed in a nylon fabric of 99.06 × 220.98 cm with a calibration range of 0–256 mmHg, 1664 detection points, and 3175 spatial resolutions per sensor, a case with housing consisting of electrical components, and a liquid crystal display monitor. Calibration is carried out by applying a known load to the sensor sheet, and the measured value is converted into millimeters of mercury (mmHg). The acquisition protocol was designed to evaluate the cushions under normal conditions, with several fixed parameters to assure test reproducibility. During each trial, the cushion was placed on the seat of subject’s own wheelchair. The subject was then seated comfortably with arms folded and feet on the footrests (which were regulated to keep the joints flexed at 90°) and maintaining an active sitting position (without putting weight on the backrest). The pelvis was placed as far back on the seat as possible with the thighs in a level position. The seat surface was horizontal and the backrest was tilted backwards by no more than 10° depending on the subject’s comfort. This posture, which was controlled with an anatomical goniometer, was defined in accordance with the previous studies [21,22,23].

During the test, the subjects remained in a sitting position with their arms crossed to the chest on the pad and their feet resting on the floor. Before recording, the participant was asked to stay on the device for 8 min, to allow them to adapt to it. After this adaptation time, the registration was carried out for 2 min. One pressure map was recorded for each subject (Figure 2). Once the pressure was recorded, the session was terminated and the recorded data were saved [24,25]. The data were shown as a color-coded map with a numerical value for each of the areas, which reflected the pressure distribution between the surface used and the user. Finally, these data were transferred to a laptop with software Pro V8 to analyze the obtained values of the variables, which were the mean (P_MEAN_), maximum (P_MAX_) and minimum (P_MIN_) pressures, and contact area of the lift base (A).

Seated Lateral Reach Test (LRT): this is the variant in sitting position of the Lateral Reach Test that evaluates the postural control of the trunk in the mid-lateral axis. It is performed by arranging the sitting person with the dominant arm extended, 90° shoulder abduction, the hand closed and the contralateral arm resting on the body. The subjects received the following standardized instructions: they had to move the arm laterally as far as possible, without losing posture control in sitting. The hips had to remain fully in contact with the surface of the pressure mapping device and no trunk flexion or rotation was allowed. The maximum perceived position was maintained for three seconds before returning to the starting position. The hand excursion was recorded laterally from the tip of the third finger. A high correlation between the LRT result and the center of body pressure was obtained [26,27].

Seated Functional Reach Test (FRT): this test evaluates the postural control of the trunk in the antero-posterior axis and has been identified as a useful test to detect the deterioration of posture control in people with disabilities. It is performed starting from a stable sitting position with the arm extended anteriorly, 90° shoulder flexion, and a closed hand [28,29].

Berg Balance Scale (BBS): this scale evaluates postural control with static and dynamic tasks in 14 sections. The tasks that are evaluated are diverse, such as getting up and sitting, sitting without support, transfers, stretching, standing without support, feet together, turning the head back and 360°, stepping up one foot on a step, etc. [30]. Although it is an evaluation initially prepared for the elderly, it has been validated for multiple populations [31].

Posture and Postural Ability Scale (PPAS): this is the modified version of the Postural Ability Scale. It evaluates sitting posture quantitatively (PPAS1) and qualitatively (PPAS2). In a quantitative way, the postural ability is cataloged in seven levels, depending on whether or not the subject is in a sitting position, with or without support, whether they are able to move, transfer weight, recover their posture, move, or even stand up. Qualitatively, it measures the position of the head, trunk, arms, pelvis, hips, knees, and feet in the frontal and sagittal plane, with a score of 0 or 1 when responding with dichotomous answers of “no” and “yes,” respectively. This test has shown excellent reliability, high internal consistency, and validity of the evaluated construct (posture quality and postural asymmetries) [32,33].

Function in Sitting Test (FIST): this test evaluates the level of functionality in sitting with 14 tasks (anterior, posterior, and lateral thrust and collection of an object that is behind, in front, next to, and at the feet of the patient, among other tasks). The total score is 56 points, with each item being assigned a score between 0 and 4 (the higher the score, the less functionality and independence) [34]. Although it was initially designed to evaluate the functional sitting posture control of people following acute stroke, its validity has been confirmed in other populations [35,36].

Trunk Control Test (TCT): it evaluates the control that the subject has over their trunk and has been correlated with the degree of functional independence, the symmetry of the center of gravity, and the Berg Balance Scale. The items that make up the scale are: rolling to the weak side, rolling to the strong side, sitting up from the lying position, and swinging in the sitting position. Each item is assigned a score in the range of 0 (unable), 12 (capable with help), and 25 (independently capable) [37,38].

### 2.3. Statistical Analysis

For the descriptive statistics, the mean was used as a measure of central tendency and the standard deviation as a measure of dispersion. The Levene test was used to demonstrate the homogeneity of the variance for all the variables from all the tests. To verify whether the differences between the groups were significant, an analysis of variance (ANOVA) test was used with the Bonferroni correction.

To determine the normality of the variables, the Shapiro-Wilk test was performed, due to the sample size (*N* less than 30). To test the hypothesis of equality of means, a single factor ANOVA was used for the variables in which the hypothesis of normal distribution (pressure, area) was accepted. The Fisher statistic was applied due to the fact that the *p-*value was higher than 0.05 for the homogeneity of the variances in the Levene’s test. To compare the groups one by one, the Tukey post-hoc test was used.

The Kurskal-Wallis non-parametric H test was used to test the equality hypothesis between the groups of variables in which the normal distribution hypothesis (P_MAX_ and P_MIN_) was not rejected.

To determine whether there was a correlation between the quantitative variables of the study, the Pearson and Spearman coefficients were used according to the type of distribution assumed for each of the variables.

We used linear regression models using the pressure mapping’ outcomes (dependent variables) and FRT and LRT (independent variables), along with adjustments for age, age of diagnosis, and sitting height. To evaluate the fit in the linear regression models, the R^2^ statistic was used. The criteria to evaluate the adjustment values higher than R^2^ > 0.25 were used as long as they were significant.

All statistical techniques were applied with STATA for MAC (version 12) and with the significance level set at *p* < 0.05.

## 3. Results

The sample consisted of 10 subjects, with 40% women and 60% men, and a mean age of 50.4 ± 7.3 years. Subjects with multiple sclerosis accounted for 20% of the sample (two women) and those with Friedreich’s ataxia (two women and two men) and spinal cord injury (four men) accounted for 40% each.

Table 1 shows the descriptive values of characterization, functional tests and pressure mapping of the total sample and by groups. The results obtained in the functional tests revealed a higher deterioration of postural control in the group of people with multiple sclerosis, who showed the lowest score (except in FIST and FRT). Higher results were observed in the Friedreich’s ataxia group with respect to the spinal cord injury group.

The pressure mapping results (Table 2 and Figure 3) showed a higher P_MEAN_ in the people with Friedreich’s ataxia (52.3 ± 3.9) compared to those with multiple sclerosis (41.7 ± 5.7) and spinal cord injury (41.3 ± 3.2). This was the only variable extracted from the pressure mapping analysis to provide statistically different results among the three groups studied. Specifically, the data showed differences in people with Friedreich’s ataxia compared to those diagnosed with multiple sclerosis (*p* = 0.04) and spinal cord injury (*p* = 0.01), with no significant differences between the latter two. However, the highest values of P_MAX_ and A were found in users with spinal cord injury, with P_MAX_ = 251.5 ± 8.9 mmHg, and A = 1146.9 ± 283.4 mmHg, although without statistically significant differences with the other two groups of pathologies. These differences can be seen in Figure 3.

The correlation analysis between the dependent variables showed significant results among all of them: P_MAX_ was correlated with A, P_MIN_ and P_MEAN_ (0.8 < *r* > 1; *p* < 0.001), A was correlated with P_MIN_ and P_MEAN_ (0.7 < *r* > 0.9; *p* < 0.001) and P_MIN_ and P_MEAN_ were also associated with each other (*r* = 0.9; *p* < 0.001). In the multiple sclerosis group, there were significant positive correlations between P_MEAN_ and LRT, FRT, and PPAS2 (0.7 < *r* > 0.9; *p* < 0.05), and a negative correlation between P_MEAN_ and TCT (*r* = −0.94; *p* < 0.001). In this group, A was also negatively correlated with LRT, FRT, and PPAS2 (−0.8 < *r* > −0.8; *p* < 0.001), and A was positively correlated with TCT (*r* = 0.9; *p* < 0.001). In the Friedreich’s ataxia group, P_MIN_ was significantly correlated with LRT and FRT (0.8 < *r* > 0.9; *p* < 0.01, for both) and A with TCT (*r* = −0.9; *p* = 0.04). Finally, in the group of people with spinal cord injury, P_MAX_ and PPAS1 were positively correlated (*r* = 0.9; *p* < 0.01).

Through a model of the linear regression for LRT and FRT, it was observed that the pressure mapping variables that most explained the results of these tests were P_MIN_ and P_MEAN_. The adjustment of the model is reflected in Table 3, and the values of R^2^ are significant (*p* < 0.05).

## 4. Discussion

The main objectives of this study were to evaluate pressure mapping in multiple sclerosis, Friedreich’s ataxia, and spinal cord injury, identifying possible differences between them, and determine the existence of relationship between the results of pressure mapping and those of the functional tests to evaluate the postural trunk control that are most widely used in rehabilitation. Once the results were analyzed, only the P_MEAN_ variable provided sufficiently different values between the three pathologies (and more sensitively than functional tests used in the clinical setting). Therefore, this variable was the only one capable of differentiating between the three subgroups of the sample. In addition, this variable shows that the behavior of sedentary people who are diagnosed with multiple sclerosis and spinal cord injury was significantly different from that of people diagnosed with Friedreich’s ataxia. Furthermore, it was observed that the highest number of relationships occurred between pressure mapping and functional tests in the multiple sclerosis group.

The reviewed literature shows that P_MEAN_ and P_MAX_ are the ones that provide more information about the distribution of pressure in sitting. Crawford et al. [39] and Stinson et al. [40] only analyzed the values of P_MEAN_ and P_MAX_, since these show the most significant differences between severely disabled people with pathologies such as multiple sclerosis, spinal cord injury, and stroke. However, this is not in line with the results obtained in this study, since P_MEAN_ and A also showed statistically significant and clinically more sensitive results than the functional tests used in the multiple sclerosis and Friedreich’s ataxia subgroups. Despite this, and in accordance with the literature published to date, P_MEAN_ is the key variable. Consequently, it could be interpreted as a fundamental parameter for the design and evaluation of specific treatments for people with these pathologies.

The protocol used to record pressure mapping consisted of 8 min of adaptation and 2 min of recording. This distribution of time is based on previous studies [25,39], which postulated that, before recording the data, the subject must spend some time on the device in order to adapt to it, verifying that the ideal duration is 8 min, and that the 2-min duration of the test means that reliable information is obtained, especially with people in wheelchairs with multiple sclerosis [39].

Regarding the functional tests, relevant differences were found in the test results by pathology and in comparison with other authors. In LRT, the normal value is 17 cm [26]. In this test, values below the normal range were obtained in the group with multiple sclerosis and spinal cord injury, which is far from the results obtained by Freeman et al. [27] and Choe et al. [29] in people with Friedreich’s ataxia, which were within the normal ranges (no background was found in the scientific literature for these results in people with this pathology).

In FRT, the normal value in the age range of 41–69 years is between 35.1–38.1 cm [28]. Similarly, values below normal were obtained in people with multiple sclerosis and spinal cord injury, which is far from the results of the studies of Frzovic et al. [41] and Kizony et al. [42]. In people with Friedreich’s ataxia, a value was achieved within the normal ranges (no background in the scientific literature was found). In FST, the total score was 56 points [34]. The data obtained in the multiple sclerosis and spinal injury groups were also below the values reported by previous studies [36,43]. Finally, in TCT, the total score was 100 points [44]. The data obtained in the multiple sclerosis and spinal injury groups were below the data collected in previous studies [45,46]. From the results obtained in the literature review of the functional tests, it is clear that the authors studied people with the same pathologies, although probably with a lower degree of disability and severity in the development of the characteristic symptomatology of these diseases.

In BBS, our populations obtained ratings between 5.5 and 15.5 points. These scores are lower than the average scores obtained by Downs et al. [31], Eftekharsadat et al. [47] (in people with multiple sclerosis), and Santos de Olivera et al. [48] (in patients with spinocerebellar ataxia). However, our results are similar to those of such studies in people with spinal cord injury. It is evident that our sample obtained lower values due to the fact that they were populations that cannot walk or stand. This condition significantly affects the score on that scale.

In the three subgroups studied, the correlations between the pressure mapping variables and the functional tests were more numerous, especially in the multiple sclerosis group. Similarly, the functional measures recorded in the same starting position (sitting) were more related to the recordings of pressure mapping in sitting. It is worth highlighting the behavior of A in LRT and FRT, which demonstrates that when the subject’s contact area in pressure mapping is lower, the scores in LRT and FRT are higher, whereas P_MEAN_ increases when the LRT and FRT scores are higher. These correlations make clear reference to the fact that the construct evaluated by pressure mapping is the state of the postural trunk control. Finally, and in line with the above, the results of the linear regression analysis support the previous findings. This analysis made it possible to identify the pressure mapping parameters that provide the most information from the LRT and FRT tests (two of the most frequently used for the assessment of postural control of the trunk in patients with wheelchairs). Their result identified the P_MEAN_ as the most important variable.

The main limitation of this work was the small sample size, which was due to the fact that they were people with a serious disability and dependence in the same care center. In the future, in addition to analyzing larger samples and patients with other neurological pathologies, it would be important to establish a correlation between the time spent in the sitting position daily and the degree of functional impairment to determine if these variables influence the results of the pressure mapping. Likewise, the novelty of the subject and the wide possibilities for future research in this field must be recognized to contribute to the development of the assistance services, which directly affects the improvement of the quality of life of people with serious disabilities and dependence. That is, the approach of specific physiotherapy protocols should be pursued taking into account the status of people with multiple sclerosis, Friedreich’s ataxia, and spinal cord injury. In addition, the study of pressures in sitting should be addressed with the use of cushions and at different angles of inclination of the back of the wheelchair to determine whether these modifications significantly reduce the pressure and if this correlates with a lower appearance of ulcers.

## 5. Conclusions

People with different neurological pathologies and similar results in functional tests have very different results when evaluated with pressure mapping. This suggests that, despite being neurological pathologies with postural control problems, they do not behave in a uniform manner and, therefore, require specific treatment approaches.

The high pressures detected by pressure mapping show the problems in the postural control of the analyzed individuals and the urgent need for measures that reduce the risk of pressure ulcers (such as the modification of the angle of inclination of the back of the wheelchair or the use of orthopedic pillows).

## Figures and Tables

**Figure 1 sensors-20-06488-f001:**
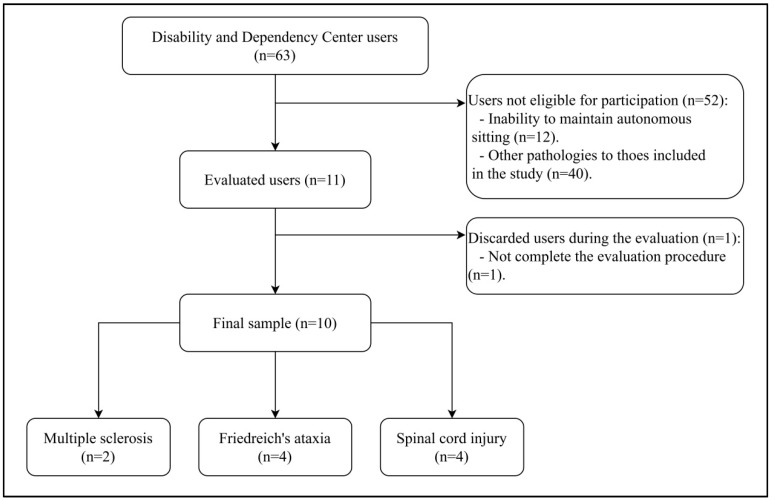
CONSORT flowchart diagram.

**Figure 2 sensors-20-06488-f002:**
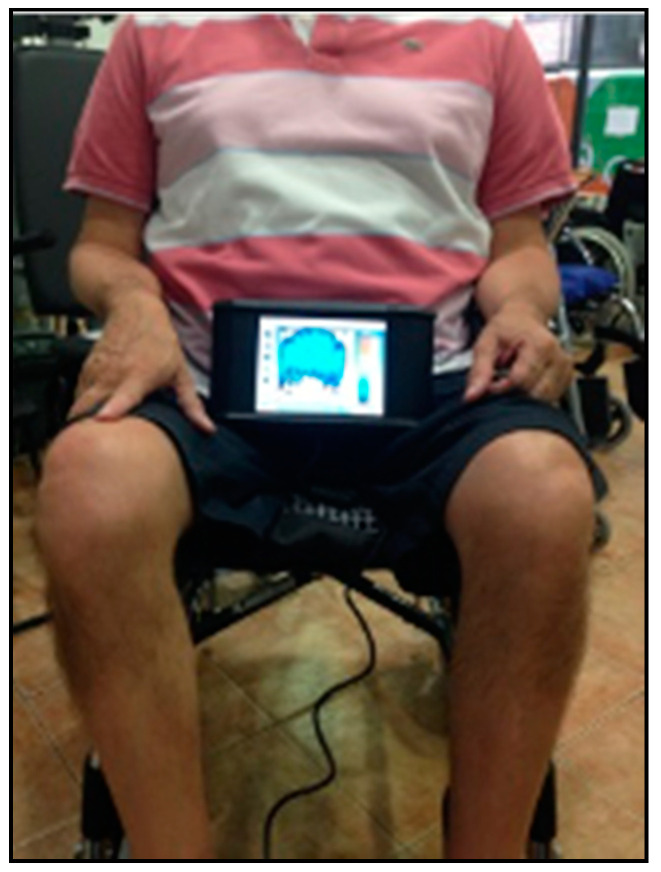
Participant during the measurement moment with pressure mapping.

**Figure 3 sensors-20-06488-f003:**
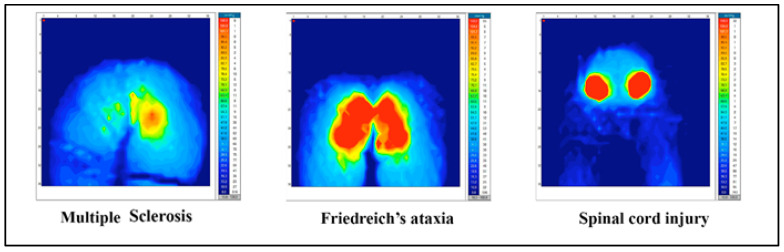
Examples of frames extracted from pressure mapping.

**Table 1 sensors-20-06488-t001:** Descriptive statistics of the sample and by groups.

	All (*n* = 10)	Multiple Sclerosis (*n* = 2)	Friedreich’s Ataxia (*n* = 4)	Spinal Cord Injury (*n* = 4)
Age (years)	49 ± 9.2	57 ± 4.2	49.3 ± 8.9	44.8 ± 10.1
Height (cm)	168.7 ± 10.7	160 ± 11.3 *^, &&&^	165.6 ± 7.9 *^, ##^	176.3 ± 9.9 ^&&&, ##^
Sitting height (cm)	80 ± 7.3	71 ± 0 ^&&&^	77 ± 3.5 ^##^	87.5 ± 2.4 ^&&&, ##^
Weight (kg)	67.8± 11.6	61.1 ± 1.3	68.1 ± 10.8	70.9 ± 15.7
BMI (kg/m^2^)	24.4 ± 3.16	24 ± 2.8	24.7 ± 1.8	24.2 ± 4.9
Duration of disease (years)	20.3 ± 1.2	19.5 ± 3.5 ***	35.2 ± 10 ***^, ##^	8.2 ± 4.8 ^##^
Previous ulcers (n)	1.6 ± 0.5	1 ± 0 *	2 ± 0 *	1.5 ± 0.6
Current ulcers (n)	1.8 ± 0.4	2 ± 0	1.8 ± 0.5	1.8 ± 0.5
MMSE (points)	29.4 ± 4.2	29 ± 4.6	29.8 ± 0.5	28.4 ± 2.6
LRT (cm)	14.7 ± 11.5	6.8 ± 1.8 **	24.9 ± 11.9 **^, ##^	8.5 ± 4.8 ^##^
FRT (cm)	24.9 ± 16.9	16.5 ± 4.9 **	38.7 ± 13.6 **^, ##^	15.3 ± 15.4 ^##^
BBS (points)	10.9 ± 7.7	5.5 ± 0.7 ^&^	9.3 ± 2.9 ^#^	15.3 ± 11.1 ^#, &^
PPAS1 (points)	9.8 ± 4	6 ± 8.5 ^&^	10.5 ± 3	11 ± 2 ^&^
PPAS2 (scale)	5.4 ± 1.2	4.5 ± 0.7	6 ± 0	5.2 ± 1.7
FIST (points)	35.5 ± 11.8	37.5 ± 3.5	42.8 ± 1.9 ^##^	27.3 ± 15.7 ^##^
TCT (points)	66.2 ± 24.1	55 ± 8.5 **^, &^	74.3 ± 20.8 **	71.3 ± 33.2 ^&^

BMI: Body Mass Index; MMSE: Mini-Mental State Examination; LRT: Seated Lateral Reach Test; FRT: Seated Functional Reach Test; BBS: Berg Balance Scale; PPAS1: Posture and Postural Ability Scale Quantitative; PPAS2: Posture and Postural Ability Scale Qualitative; FIST: Function in Sitting Test; TCT: Trunk Control Test. Significant comparison between Multiple Sclerosis vs. Friedreich’s ataxia: * *p* < 0.05; ** *p* < 0.01; *** *p* < 0.001. Significant comparison between Multiple Sclerosis vs. Spinal cord injury: ^&^
*p* < 0.05; ^&&&^
*p* < 0.001. Significant comparison between Friedreich’s ataxia vs. Spinal cord injury: ^##^
*p* < 0.01.

**Table 2 sensors-20-06488-t002:** Results of the blanket of pressures by groups and comparison between them.

	All (*n* = 10)	Multiple Sclerosis (*n* = 2)	Friedreich’s Ataxia (*n* = 4)	Spinal Cord Injury (*n* = 4)
P_MEAN_ (mmHg)	45.8 ± 6.6	41.7 ± 5.7 *	52.3 ± 3.9 *^, ##^	41.3 ± 3.2 ^##^
P_MAX_ (mmHg)	239.4 ± 33.6	204.7 ± 72.6	244.5 ± 22.9	251.5 ± 8.9
P_MIN_ (mmHg)	10.2 ± 0.2	10.1 ± 0.1	10.3 ± 0.2	10.1 ± 0.9
A (cm^2^)	1095.3 ± 208.5	1036.7 ± 310.8	1073 ± 106.7	1146.9 ± 283.4

P_MEAN_: Mean pressure; P_MAX_: Maximum pressure; P_MIN_: Minimum pressure; A: contact area. Significant comparison between Multiple Sclerosis vs. Friedreich’s ataxia: * *p* < 0.05. Significant comparison between Friedreich’s ataxia vs. Spinal cord injury: ^##^
*p* < 0.01.

**Table 3 sensors-20-06488-t003:** Linear regression models for the Seated Lateral Reach Test and Seated Functional Reach Test (continuous variables).

Variables Included	Seated Lateral Reach Test			Seated Functional Reach Test		
	B	SE	R^2^	B	SE	R^2^
P_MEAN_	1.13 *	0.471	0.25 ***	1.82 *	0.637	0.29 **
P_MAX_	0.03	0.12	0.025 *	0.08	0.175	0.26*
P_MIN_	53.1 **	14.788	0.32 ***	82.24 ***	19.588	0.33 ***
A	−0.02	0.019	0.26 **	−0.02	0.028	0.26 *

B—regression coefficient; SE—standard error; R^2^: coefficient of determination.*: *p*-value < 0.05; **: *p*-value < 0.01; ***: *p*-value < 0.001.

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
