# Peer review of "Analysis of Postural Control in Sitting by Pressure Mapping in Patients with Multiple Sclerosis, Spinal Cord Injury and Friedreich’s Ataxia: A Case Series Study"

_sensors, 2020, doi:10.3390/s20226488_

Round 1
Reviewer 1 Report
I have read a previous version of this manuscript by Reguera-García and colleagues. The manuscript has now improved but still some of my comments have remained unattended. Also, I believe this sample size would be much appropriate for a case series, rather than a full paper.
In the general description of the technique, authors mention “this system evaluates the contact surface of the subjects in a wheelchair, thus providing information on how their positioning is”. As such, it remains unclear whether this study is limited to wheelchair bound patients. If so, authors can definitely discuss on posture, but need to remove any reference to balance (which by definition requires to stand).
There is another my previous comment that has gone unattended. Looking at figure 2, it seems that spinal cord injury and Friedreich ataxia have a symmetric pattern of pressure mapping, whilst multiple sclerosis has an asymmetric pattern. Was this actually observed by the authors? It would be interesting to know the pattern of symptoms (unilateral vs bilateral).
Authors continue on reporting p-values for multiple statistical tests. In my previous comments, I suggested the authors provided readers with more descriptive results, considering that statistical tests do not have high reliability on this sample size. Since the authors do not seem to have appreciated this suggestion, at this point, I request correction for multiple comparisons. Comments should then only limited to statistically significant results.
The discussion remains too long. I would focus on the relevance of pressure mapping, when compared with other tests, and on the actual novelties of this study. Also, I would avoid any speculation, which are definitely not allowed in consideration of the sample size.
Author Response
Dear Editor and Reviewer of Sensors:
Thank you very much for your suggestions and contributions to improve the quality of the manuscript. Following your indications, we respond, point by point, to the reviewers' comments.
In the text, all the modified or added sentences have been written in red to facilitate the correction by the reviewers.
- I have read a previous version of this manuscript by Reguera-García and colleagues. The manuscript has now improved but still some of my comments have remained unattended. Also, I believe this sample size would be much appropriate for a case series, rather than a full paper.
The type of study has been modified to case series study.
- In the general description of the technique, authors mention “this system evaluates the contact surface of the subjects in a wheelchair, thus providing information on how their positioning is”. As such, it remains unclear whether this study is limited to wheelchair bound patients. If so, authors can definitely discuss on posture, but need to remove any reference to balance (which by definition requires to stand).
The term "balance" has been removed from the entire manuscript and replaced by "postural control".
At the end of the Introduction, in the definition of the objective it was specified that the study was carried out with patients in wheelchairs.
- Looking at figure 2, it seems that spinal cord injury and Friedreich ataxia have a symmetric pattern of pressure mapping, whilst multiple sclerosis has an asymmetric pattern. Was this actually observed by the authors? It would be interesting to know the pattern of symptoms (unilateral vs bilateral).
Your appreciation of Figure 3 is correct but the reviewer must take into account that in said figure the images included are EXAMPLES of three patients (NOT REPRESENTATIVE OF ALL PATIENTS WITH THE SAME PATHOLOGIES).
In other words, the other patients with Multiple Sclerosis did not show this asymmetric pattern. If so, the reviewer can be sure that the authors would have transmitted it in the manuscript.
- Authors continue on reporting p-values for multiple statistical tests. In my previous comments, I suggested the authors provided readers with more descriptive results, considering that statistical tests do not have high reliability on this sample size. Since the authors do not seem to have appreciated this suggestion, at this point, I request correction for multiple comparisons. Comments should then only limited to statistically significant results.
The Results section has been summarized and Table 3 has been removed (in the previous version of the manuscript there were four tables)
- The discussion remains too long. I would focus on the relevance of pressure mapping, when compared with other tests, and on the actual novelties of this study. Also, I would avoid any speculation, which are definitely not allowed in consideration of the sample size.
The discussion has been summarized. In addition, some expressions have been eliminated or reformulated to avoid transmitting speculation not supported by the results.
Once again, thank you very much for the time spent and the interest shown in this work; as well as in the positive evaluations you have given of it.
Receive a warm greeting,
The authors.
Reviewer 2 Report
The authors present a very interesting study investigating differences in sitting pressure maps during functional activities in subjects with MS, spinal cord injury, and Friedreich's ataxia. The number of subjects is low, however the topic is very intriguing and despite the low number of subjects, the authors found statistically significant differences between groups (specifically between subjects with Friedreich's ataxia compared with the subjects who had MS or spinal cord injury). The methodology is clear and sufficient for capturing the biomechanical pressure metrics, and the discussion and conclusions are consistent with the reported results.
Minor comments:
- although the changes in pressure mapping are clearly seen, the connection with deteriorated motor control is less clear. Based on Table 1, the highest deterioration in postural control was in the MS group, yet the Friedreich's ataxia group was the only significantly different change in pressure mapping.
- Of the three research questions and three hypotheses put forth in the introduction (lines 73-81), only one research question and one hypothesis was addressed in a meaningful way. I suggest re-focusing the statement of purpose to align with what was addressed.
Author Response
Dear Editor and Reviewer of Sensors:
Thank you very much for your suggestions and contributions to improve the quality of the manuscript. Following your indications, we respond, point by point, to the reviewers' comments.
In the text, all the modified or added sentences have been written in red to facilitate the correction by the reviewers.
- The authors present a very interesting study investigating differences in sitting pressure maps during functional activities in subjects with MS, spinal cord injury, and Friedreich's ataxia. The number of subjects is low, however the topic is very intriguing and despite the low number of subjects, the authors found statistically significant differences between groups (specifically between subjects with Friedreich's ataxia compared with the subjects who had MS or spinal cord injury). The methodology is clear and sufficient for capturing the biomechanical pressure metrics, and the discussion and conclusions are consistent with the reported results.
The authors appreciate the very positive assessment of our research.
- Although the changes in pressure mapping are clearly seen, the connection with deteriorated motor control is less clear. Based on Table 1, the highest deterioration in postural control was in the MS group, yet the Friedreich's ataxia group was the only significantly different change in pressure mapping. Of the three research questions and three hypotheses put forth in the introduction (lines 73-81), only one research question and one hypothesis was addressed in a meaningful way. I suggest re-focusing the statement of purpose to align with what was addressed.
Taking into account the small sample size used, the authors have reduced the study objectives to the first two that we had proposed (and we have eliminated the third of them). Simplifying the approach of the study in Introduction but also clarifying the meaning of the data obtained.
Once again, thank you very much for the time spent and the interest shown in this work; as well as in the positive evaluations you have given of it.
Receive a warm greeting,
The authors.
Reviewer 3 Report
This study indicated that he relation between the pressure mapping variables and the functional tests were numerous, especially in the multiple sclerosis group. This study provides new information in posture control for clinical staffs. A revision is suggested.
1.Please update new studies which are related posture control ( sitting ) in neurological patients.
- Please address how the N number was calculated.
- Please emphasize the clinical implications of this study.
Author Response
Dear Editor and Reviewer of Sensors:
Thank you very much for your suggestions and contributions to improve the quality of the manuscript. Following your indications, we respond, point by point, to the reviewers' comments.
In the text, all the modified or added sentences have been written in red to facilitate the correction by the reviewers.
- Please update new studies which are related posture control ( sitting ) in neurological patients.
The authors have added four new bibliographic references (three of them published in the last months).
- Please address how the N number was calculated.
The sample is the result of a non-probabilistic process of convenience (that is, all possible patients who wanted to participate and who fulfilled the criteria were recruited without having a calculated N number as a target).
This aspect has been conveyed in the manuscript.
- Please emphasize the clinical implications of this study.
The authors have expanded the last paragraph of the Discussion and Conclusions to include this aspect.
Once again, thank you very much for the time spent and the interest shown in this work; as well as in the positive evaluations you have given of it.
Receive a warm greeting,
The authors.
Reviewer 4 Report
The present study presents a very interesting and powerful evaluation of a small series of neurological patients. However, the objectives and type of study are not consistent with the small sample size and the subgroups created. Some of the clinical and methodological recommendations that I make to you could be taken into account in future research.
Page 1. Line 3. It would be interesting to include the type of study in the title.
Page 1. Line 23. More sensitive measuring what variable?
Page 1. Line 24. This type of study is carried out with a representative number of subjects. In this study, only a series of cases divided into subgroups have been taken into account.
Page 1. 28-30. “A higher mean pressure” and “highest values of maximun pressure and the area of contact”. What variable is being evaluated by these measurements?
Page 1. 32. It is not possible to establish a relationship with such a small sample size.
Page 1. 34 “Measurement equipment”. This is not a MeSH term. Could it be replaced by "Analytical, Diagnostic and Therapeutic Techniques and Equipment Category"?
Page 1. Line 39. “A coordinated movement are coordinated”. It would be necessary to improve the expression.
Page 2. Line 44. References 1,2 (There are much more recent publications on theories of motor control. Would it be possible to replace these references?).
Page 2. Line 48. References 3-6 (Please check if these references would be the most suitable for these statements).
Page 2. Line 54. References 9,10 (These references are focused on age-related motor control disorders, not neurological disorders. It would be important to replace them).
Page 2. Line 56. References 11,12 (References focused on "older adults" not on “neurological disorders”).
Page 2. Line 55-56. Include references.
Page 2. Line 57-58. To assess postural control, there are validated portable devices that can be used in clinical evaluation outside the laboratory. Please modify.
Page 2. Line 77. b) Please, include the variable in which you want to determine the sensitivity.
Page 2. Line 78-79. c) To meet this objective, a statistical correlation would be necessary, which, given the small number of subjects, would not be possible.
Page 2. Line 84. “random” (The randomization of the subjects is not the way in which patients were included, since patients are included in each of the three groups based on their pathology. Would it be possible to remove this term?)
Page 2. Line 86. Are you referring to institutionalized or outpatient patients?
Page 2. Line 89. Why were other neurological pathologies that also have this deficit in postural control not included in an effort to increase the sample size?
Page 3. Line 95. The initial N of subjects should include, at least, the subjects that present neurological pathology, specify what other neurological pathologies are treated and why they were not included.
Page 3. Line 108. “weight, height, sitting height” (These would be more anthropometric than socio-demographic data).
Page 3. Line 109. Why weren't "pressure ulcer risk scales" like the Norton Scale used?
Page 3. Line 113. References 16,17 (Would it be possible to include references on the validation of this device in neurological pathologies?)
Page 4. Line 123. If the patient's trunk was supported by the backrest, how could it be determined if the amount of pressure exerted on the backrest did not modify the distribution of pressure exerted on the registered seat surface?
If among the inclusion criteria, the patient had to remain without backup support, why was the test recorded with backup support?
Page 4. Line 154. The clinical tests were implemented by the same physiotherapist who performed the instrumentalized evaluation?
Page 5. Line 181. References 30,31 (These references focus on "Cerebral Palsy". Could you provide references that validate this scale in adult neurological pathology?)
Page 5. Line 193. Reference 34. Could you provide references of its validation in the neurological pathologies included in the study?
Page 5. Line 201. N = 2 in the subgroups is an excessively small sample size, even taking into account the choice of this test for N <30.
Page 5. Line 208. The statistical analysis has been very detailed. However, a statistical analysis prior to the one that has been detailed is necessary to calculate the minimum sample size to confirm that the statistical correlation established is valid. Without this first analysis, the rest of the statistics would be invalidated, since with groups of 2 subjects, the general behavior that patients diagnosed with this pathology would have cannot be inferred.
Page 7. Line 208. Could the representation of an asymmetric distribution between the two hemibodies be due to the greater involvement of one hemibody that these patients have? Could this be the difference between this pathology and the rest? In this sense, it would be interesting to include pathologies such as stroke that occur predominantly with this asymmetry.
Page 8. Line 284. Reference 35 (What explanation could you give to this similarity between "sledge hockey players" and patients with neurological pathology?)
Page 8. Line 289. Reference 36 (It would be important to establish a correlation between the time spent in the sitting position daily, which could greatly affect the results of the present study. It would be interesting to determine how much time the patients spent sitting in the wheelchair on average to determine if the possible changes between patients could be more due to differences in this temporality than to the type of pathology).
Page 8. Line 305. In the results, the "duration of disease" has been taken into account but not the functional impairment of the patients. Patients with severe affectation, determined for example by the score in the EDDS in multiple sclerosis, could be determining the functional differences found in this pathology, not being able to extrapolate the results to the pathology in a global way.
Page 9. Line 332. It would be necessary to limit the functional stage of the groups using a specific range of scores of some of the scales as a screening.
Page 9. Line 334. For strong statistical correlations to exist, the sample size has to be significantly larger.
Page 9. Line 334-335. Does this mean then that there were no differences between clinical tests and instrumentalized evaluation? If there was a strong correlation, this would be in contradiction with the Conclusions presented.
Author Response
Dear Editor and Reviewer of Sensors:
Thank you very much for your suggestions and contributions to improve the quality of the manuscript. Following your indications, we respond, point by point, to the reviewers' comments.
In the text, all the modified or added sentences have been written in red to facilitate the correction by the reviewers.
- Page 1. Line 3. It would be interesting to include the type of study in the title.
The authors have modified the title following your advice.
- Page 1. Line 23. More sensitive measuring what variable?
The authors wanted to refer to "the variables extracted from the pressure mapping analysis". The sentence has been corrected.
- Page 1. Line 24. This type of study is carried out with a representative number of subjects. In this study, only a series of cases divided into subgroups have been taken into account.
That aspect of the methodology has been corrected.
- Page 1. 28-30. “A higher mean pressure” and “highest values of maximun pressure and the area of contact”. What variable is being evaluated by these measurements?
The abbreviations of the variables to which we refer have been included.
- Page 1. 32. It is not possible to establish a relationship with such a small sample size.
The term "relation" has been changed to "correlation" which is more correct for this sentence.
- Page 1. 34 “Measurement equipment”. This is not a MeSH term. Could it be replaced by "Analytical, Diagnostic and Therapeutic Techniques and Equipment Category"?
The keyword has been replaced by the one you recommend.
- Page 1. Line 39. “A coordinated movement are coordinated”. It would be necessary to improve the expression.
The authors regret the mistake (which we have already corrected).
- Page 2. Line 44. References 1,2 (There are much more recent publications on theories of motor control. Would it be possible to replace these references?).
The authors have replaced the references with three more recent ones that agree with the information we want to convey.
- Page 2. Line 48. References 3-6 (Please check if these references would be the most suitable for these statements).
The authors have replaced the references with two more appropriate ones.
- Page 2. Line 54. References 9,10 (These references are focused on age-related motor control disorders, not neurological disorders. It would be important to replace them).
The authors have replaced the references with two more appropriate ones.
- Page 2. Line 56. References 11,12 (References focused on "older adults" not on “neurological disorders”).
The authors have replaced the references with two more appropriate ones.
- Page 2. Line 55-56. Include references.
The authors have added a reference that conveys that information.
- Page 2. Line 57-58. To assess postural control, there are validated portable devices that can be used in clinical evaluation outside the laboratory. Please modify.
The authors wanted to refer to the tests and functional test scales. The phrase has been corrected.
- Page 2. Line 77. Please, include the variable in which you want to determine the sensitivity.
The research question has been reformulated.
- Page 2. Line 78-79. c) To meet this objective, a statistical correlation would be necessary, which, given the small number of subjects, would not be possible.
That objective has been removed.
- Page 2. Line 84. “random” (The randomization of the subjects is not the way in which patients were included, since patients are included in each of the three groups based on their pathology. Would it be possible to remove this term?)
The term has been removed from the body of the manuscript (and from the Abstract where it also appeared).
- Page 2. Line 86. Are you referring to institutionalized or outpatient patients?
The center serves patients in both modes (this detail has been added in the text).
- Page 2. Line 89. Why were other neurological pathologies that also have this deficit in postural control not included in an effort to increase the sample size?
They were the three specific diagnoses that had been identified in the center's registry (other patients had nonspecific syndromes). Since the lack of a specific diagnosis is a strange variable, the authors defined said inclusion criteria.
- Page 3. Line 95. The initial N of subjects should include, at least, the subjects that present neurological pathology, specify what other neurological pathologies are treated and why they were not included.
The initial N appears in the third line of that paragraph (and in Figure 1).
The other pathologies were non-specific syndromes or had multiple pathologies, so for this study we preferred not to include them.
- Page 3. Line 108. “weight, height, sitting height” (These would be more anthropometric than socio-demographic data).
The authors have included the term you refer to.
- Page 3. Line 109. Why weren't "pressure ulcer risk scales" like the Norton Scale used?
Because the main objective was to compare clinical tests of postural control with pressure mapping (which already implies the use of many evaluation instruments). The instrument you recommend is of great interest and we the authors will take it into account in our future research.
- Page 3. Line 113. References 16,17 (Would it be possible to include references on the validation of this device in neurological pathologies?)
The authors have included a new reference that analyzes this device.
- Page 4. Line 123. If the patient's trunk was supported by the backrest, how could it be determined if the amount of pressure exerted on the backrest did not modify the distribution of pressure exerted on the registered seat surface?
The evaluating physiotherapist gave the order to CONTACT the backrest to ensure that all patients were supported to the bottom of the pressure mapping. But they were also ordered to maintain an active sitting position (without putting weight on the backrest).
This detail has been added to the description of the procedure in the manuscript.
- If among the inclusion criteria, the patient had to remain without backup support, why was the test recorded with backup support?
The inclusion criteria was necessary to be able to perform all evaluation tests with confidence.
As in some wheelchairs the backrest was not removable, the authors decided not to remove it in order to homogenize the measurements (in addition, the backrest was a CONTACT, not weight-bearing).
- Page 4. Line 154. The clinical tests were implemented by the same physiotherapist who performed the instrumentalized evaluation?
Yes, the entire evaluation procedure was performed by the same physiotherapist (previously trained and trained for it).
This detail has been specified in the manuscript.
- Page 5. Line 181. References 30,31 (These references focus on "Cerebral Palsy". Could you provide references that validate this scale in adult neurological pathology?)
Only one reference has been found that evaluates this Scale in adults (and it has been included).
- Page 5. Line 193. Reference 34. Could you provide references of its validation in the neurological pathologies included in the study?
Only one reference has been found that evaluates this Scale in adults (and it has been included).
- Page 5. Line 201. N = 2 in the subgroups is an excessively small sample size, even taking into account the choice of this test for N < 30.
The authors take into account the limitations in the interpretation of our results and this is reflected in the limitations of the research (at the end of the Discussion).
- Page 7. Line 208. Could the representation of an asymmetric distribution between the two hemibodies be due to the greater involvement of one hemibody that these patients have? Could this be the difference between this pathology and the rest? In this sense, it would be interesting to include pathologies such as stroke that occur predominantly with this asymmetry.
Your appreciation of Figure 3 is correct but the reviewer must take into account that in said figure the images included are EXAMPLES of three patients (NOT REPRESENTATIVE OF ALL PATIENTS WITH THE SAME PATHOLOGIES).
In other words, the other patients with Multiple Sclerosis did not show this asymmetric pattern. If so, the reviewer can be sure that the authors would have transmitted it in the manuscript.
- Page 8. Line 284. Reference 35 (What explanation could you give to this similarity between "sledge hockey players" and patients with neurological pathology?)
The authors recognize that the choice of this bibliographic reference is unwise. That reference has been removed. Also, another reviewer asked us to summarize the Discussion so that paragraph was removed.
- Page 8. Line 289. Reference 36 (It would be important to establish a correlation between the time spent in the sitting position daily, which could greatly affect the results of the present study. It would be interesting to determine how much time the patients spent sitting in the wheelchair on average to determine if the possible changes between patients could be more due to differences in this temporality than to the type of pathology).
The variable you indicate has been included as a suggestion for improvement in future research.
- Page 8. Line 305. In the results, the "duration of disease" has been taken into account but not the functional impairment of the patients. Patients with severe affectation, determined for example by the score in the EDDS in multiple sclerosis, could be determining the functional differences found in this pathology, not being able to extrapolate the results to the pathology in a global way.
The variable you indicate has been included as a suggestion for improvement in future research.
Please bear in mind that in this research we already used 20 study variables. The instrument you recommend would undoubtedly be of great interest, but it is not essential to achieve the objectives of this research.
- Page 9. Line 332. It would be necessary to limit the functional stage of the groups using a specific range of scores of some of the scales as a screening.
The authors have rewritten the sentence avoiding inappropriate adjectives.
- Page 9. Line 334. For strong statistical correlations to exist, the sample size has to be significantly larger.
The authors have rewritten the sentence avoiding inappropriate adjectives.
- Page 9. Line 334-335. Does this mean then that there were no differences between clinical tests and instrumentalized evaluation? If there was a strong correlation, this would be in contradiction with the Conclusions presented.
The authors have rewritten that sentence. The correlation is high but, at the same time, pressure mapping provides more sensitive information (different scores between patients who have similar results in functional tests).
Once again, thank you very much for the time spent and the interest shown in this work; as well as in the positive evaluations you have given of it.
Receive a warm greeting,
The authors.
Round 2
Reviewer 1 Report
Thanks for addressing my concerns.
Author Response
Dear Editor,
Thank you very much for your positive assessment of our work.
Kind regards.
The authors.
Reviewer 4 Report
Dear authors,
thanks for the effort made. I leave you some final observations, emphasizing that it is a preliminary study without statistical data that can be extrapolated. Congratulations on the work done.
Dear Editor and Reviewer of Sensors:
Thank you very much for your suggestions and contributions to improve the quality of the manuscript. Following your indications, we respond, point by point, to the reviewers' comments.
In the text, all the modified or added sentences have been written in red to facilitate the correction by the reviewers.
- Page 1. Line 3. It would be interesting to include the type of study in the title.
The authors have modified the title following your advice.
Reviewer: Thank you for taking my recommendation into account, the type of study offers important information. Since the three pathologies only share some clinical manifestations, perhaps the title "Analysis of postural control in sitting by pressure mapping in patients with multiple sclerosis, spinal cord injury and Friedreich's ataxia: A case series study" could be replaced by "Analysis of postural control in sitting by pressure mapping in different neurological diseases. A case series study "
- Page 1. Line 23. More sensitive measuring what variable?
The authors wanted to refer to "the variables extracted from the pressure mapping analysis". The sentence has been corrected.
Reviewer: Sorry, I may not have clarified well. As for the variable, you should collect specific variables where these measurements are more sensitive than functional tests (for example, on page 9, line 335, it refers to "Postural Trunk Control"). Could you group all the functional tests in this same variable? If so, the sentence could be " the variables extracted from the pressure mapping analysis " and at the end of the sentence add the variable where it is most sensitive.
- Page 1. Line 24. This type of study is carried out with a representative number of subjects. In this study, only a series of cases divided into subgroups have been taken into account.
That aspect of the methodology has been corrected.
- Page 1. 28-30. “A higher mean pressure” and “highest values of maximun pressure and the area of contact”. What variable is being evaluated by these measurements?
The abbreviations of the variables to which we refer have been included.
Reviewer: I was not referring to abbreviations. They are data from a measuring instrument. But what variable do they evaluate?
- Page 1. 32. It is not possible to establish a relationship with such a small sample size.
The term "relation" has been changed to "correlation" which is more correct for this sentence.
Reviewer: It could be replaced by: "Although it is not possible to establish a strong statistical correlation, the relationship between .... seems…”
- Page 1. 34 “Measurement equipment”. This is not a MeSH term. Could it be replaced by "Analytical, Diagnostic and Therapeutic Techniques and Equipment Category"?
The keyword has been replaced by the one you recommend.
Reviewer: Sorry, the appropriate keyword was”Diagnostic Equipment”.
- Page 1. Line 39. “A coordinated movement are coordinated”. It would be necessary to improve the expression.
The authors regret the mistake (which we have already corrected).
- Page 2. Line 44. References 1,2 (There are much more recent publications on theories of motor control. Would it be possible to replace these references?).
The authors have replaced the references with three more recent ones that agree with the information we want to convey.
- Page 2. Line 48. References 3-6 (Please check if these references would be the most suitable for these statements).
The authors have replaced the references with two more appropriate ones.
- Page 2. Line 54. References 9,10 (These references are focused on age-related motor control disorders, not neurological disorders. It would be important to replace them).
The authors have replaced the references with two more appropriate ones.
- Page 2. Line 56. References 11,12 (References focused on "older adults" not on “neurological disorders”).
The authors have replaced the references with two more appropriate ones.
- Page 2. Line 55-56. Include references.
The authors have added a reference that conveys that information.
- Page 2. Line 57-58. To assess postural control, there are validated portable devices that can be used in clinical evaluation outside the laboratory. Please modify.
The authors wanted to refer to the tests and functional test scales. The phrase has been corrected.
- Page 2. Line 77. Please, include the variable in which you want to determine the sensitivity.
The research question has been reformulated.
Reviewer: The indications already mentioned in the Abstract can be followed.
- Page 2. Line 78-79. c) To meet this objective, a statistical correlation would be necessary, which, given the small number of subjects, would not be possible.
That objective has been removed.
- Page 2. Line 84. “random” (The randomization of the subjects is not the way in which patients were included, since patients are included in each of the three groups based on their pathology. Would it be possible to remove this term?)
The term has been removed from the body of the manuscript (and from the Abstract where it also appeared).
- Page 2. Line 86. Are you referring to institutionalized or outpatient patients?
The center serves patients in both modes (this detail has been added in the text).
- Page 2. Line 89. Why were other neurological pathologies that also have this deficit in postural control not included in an effort to increase the sample size?
They were the three specific diagnoses that had been identified in the center's registry (other patients had nonspecific syndromes). Since the lack of a specific diagnosis is a strange variable, the authors defined said inclusion criteria.
- Page 3. Line 95. The initial N of subjects should include, at least, the subjects that present neurological pathology, specify what other neurological pathologies are treated and why they were not included.
The initial N appears in the third line of that paragraph (and in Figure 1).
The other pathologies were non-specific syndromes or had multiple pathologies, so for this study we preferred not to include them.
- Page 3. Line 108. “weight, height, sitting height” (These would be more anthropometric than socio-demographic data).
The authors have included the term you refer to.
- Page 3. Line 109. Why weren't "pressure ulcer risk scales" like the Norton Scale used?
Because the main objective was to compare clinical tests of postural control with pressure mapping (which already implies the use of many evaluation instruments). The instrument you recommend is of great interest and we the authors will take it into account in our future research.
- Page 3. Line 113. References 16,17 (Would it be possible to include references on the validation of this device in neurological pathologies?)
The authors have included a new reference that analyzes this device.
- Page 4. Line 123. If the patient's trunk was supported by the backrest, how could it be determined if the amount of pressure exerted on the backrest did not modify the distribution of pressure exerted on the registered seat surface?
The evaluating physiotherapist gave the order to CONTACT the backrest to ensure that all patients were supported to the bottom of the pressure mapping. But they were also ordered to maintain an active sitting position (without putting weight on the backrest).
This detail has been added to the description of the procedure in the manuscript.
- If among the inclusion criteria, the patient had to remain without backup support, why was the test recorded with backup support?
The inclusion criteria was necessary to be able to perform all evaluation tests with confidence.
As in some wheelchairs the backrest was not removable, the authors decided not to remove it in order to homogenize the measurements (in addition, the backrest was a CONTACT, not weight-bearing).
- Page 4. Line 154. The clinical tests were implemented by the same physiotherapist who performed the instrumentalized evaluation?
Yes, the entire evaluation procedure was performed by the same physiotherapist (previously trained and trained for it).
This detail has been specified in the manuscript.
- Page 5. Line 181. References 30,31 (These references focus on "Cerebral Palsy". Could you provide references that validate this scale in adult neurological pathology?)
Only one reference has been found that evaluates this Scale in adults (and it has been included).
- Page 5. Line 193. Reference 34. Could you provide references of its validation in the neurological pathologies included in the study?
Only one reference has been found that evaluates this Scale in adults (and it has been included).
- Page 5. Line 201. N = 2 in the subgroups is an excessively small sample size, even taking into account the choice of this test for N < 30.
The authors take into account the limitations in the interpretation of our results and this is reflected in the limitations of the research (at the end of the Discussion).
- Page 7. Line 208. Could the representation of an asymmetric distribution between the two hemibodies be due to the greater involvement of one hemibody that these patients have? Could this be the difference between this pathology and the rest? In this sense, it would be interesting to include pathologies such as stroke that occur predominantly with this asymmetry.
Your appreciation of Figure 3 is correct but the reviewer must take into account that in said figure the images included are EXAMPLES of three patients (NOT REPRESENTATIVE OF ALL PATIENTS WITH THE SAME PATHOLOGIES).
Reviewer: Normally, when images are included as examples, they are usually representative of group behavior.
In other words, the other patients with Multiple Sclerosis did not show this asymmetric pattern. If so, the reviewer can be sure that the authors would have transmitted it in the manuscript.
Reviewer: The reviewer doubted the representativeness of this pattern because he did not find Figure 3 referenced in the manuscript (all tables and figures have to be referenced in the text). Also replace "Sclerosis Multiple " with "Multiple Sclerosis" in the image.
- Page 8. Line 284. Reference 35 (What explanation could you give to this similarity between "sledge hockey players" and patients with neurological pathology?)
The authors recognize that the choice of this bibliographic reference is unwise. That reference has been removed. Also, another reviewer asked us to summarize the Discussion so that paragraph was removed.
- Page 8. Line 289. Reference 36 (It would be important to establish a correlation between the time spent in the sitting position daily, which could greatly affect the results of the present study. It would be interesting to determine how much time the patients spent sitting in the wheelchair on average to determine if the possible changes between patients could be more due to differences in this temporality than to the type of pathology).
The variable you indicate has been included as a suggestion for improvement in future research.
- Page 8. Line 305. In the results, the "duration of disease" has been taken into account but not the functional impairment of the patients. Patients with severe affectation, determined for example by the score in the EDDS in multiple sclerosis, could be determining the functional differences found in this pathology, not being able to extrapolate the results to the pathology in a global way.
The variable you indicate has been included as a suggestion for improvement in future research.
Please bear in mind that in this research we already used 20 study variables. The instrument you recommend would undoubtedly be of great interest, but it is not essential to achieve the objectives of this research.
Reviewer: Variables on the functional stage are essential in studies evaluating motor control in degenerative neurological pathologies. Post-diagnosis time can be an interesting variable in stroke, for example, that occurs suddenly, in degenerative pathologies a functional assessment is necessary because there is not always a correlation in post-diagnosis time and functional stage. In your case, postural control will depend directly on postural control. Please keep that in mind.
- Page 9. Line 332. It would be necessary to limit the functional stage of the groups using a specific range of scores of some of the scales as a screening.
The authors have rewritten the sentence avoiding inappropriate adjectives.
- Page 9. Line 334. For strong statistical correlations to exist, the sample size has to be significantly larger.
The authors have rewritten the sentence avoiding inappropriate adjectives.
- Page 9. Line 334-335. Does this mean then that there were no differences between clinical tests and instrumentalized evaluation? If there was a strong correlation, this would be in contradiction with the Conclusions presented.
The authors have rewritten that sentence. The correlation is high but, at the same time, pressure mapping provides more sensitive information (different scores between patients who have similar results in functional tests).
Once again, thank you very much for the time spent and the interest shown in this work; as well as in the positive evaluations you have given of it.
Receive a warm greeting,
The authors.
Author Response
Dear Editor and Reviewer of Sensors:
Thank you very much for your suggestions and contributions to improve the quality of the manuscript. Following your indications, we respond, point by point, to the reviewers' comments.
In the text, all the modified or added sentences have been written in red to facilitate the correction by the reviewers.
- Thank you for taking my recommendation into account, the type of study offers important information. Since the three pathologies only share some clinical manifestations, perhaps the title "Analysis of postural control in sitting by pressure mapping in patients with multiple sclerosis, spinal cord injury and Friedreich's ataxia: A case series study" could be replaced by "Analysis of postural control in sitting by pressure mapping in different neurological diseases. A case series study"
Thank you very much for your recommendation to improve the title. The authors are aware of the importance of the correct choice of the title of a manuscript and we have modified the title according to your suggestion.
- Sorry, I may not have clarified well. As for the variable, you should collect specific variables where these measurements are more sensitive than functional tests (for example, on page 9, line 335, it refers to "Postural Trunk Control"). Could you group all the functional tests in this same variable? If so, the sentence could be " the variables extracted from the pressure mapping analysis " and at the end of the sentence add the variable where it is most sensitive.
We, the authors, did not quite understand your correction in the first letter. Now that detail has been corrected (in the Abstract, Introduction and Discussion).
- I was not referring to abbreviations. They are data from a measuring instrument. But what variable do they evaluate?
The authors, again, we had not understood the correction you were making us.
We have modified the phrase: "a higher mean pressure on the seat of subject's wheelchair compared to other groups”.
- It could be replaced by: "Although it is not possible to establish a strong statistical correlation, the relationship between .... seems…”
The authors have corrected the sentence following your instructions.
- Sorry, the appropriate keyword was”Diagnostic Equipment”.
The key word has been corrected.
- The indications already mentioned in the Abstract can be followed.
This sentence has also been corrected.
- Normally, when images are included as examples, they are usually representative of group behavior.
No representative pattern was found (due to the small sample size). Despite this, the authors believe it is important to include the images because they are different from each other and, as pressure mapping is a rare instrument, readers can know the resulting image.
- The reviewer doubted the representativeness of this pattern because he did not find Figure 3 referenced in the manuscript (all tables and figures have to be referenced in the text). Also replace "Sclerosis Multiple " with "Multiple Sclerosis" in the image.
The authors have corrected the references to Figure 3 in the text and its legend has been corrected.
- Variables on the functional stage are essential in studies evaluating motor control in degenerative neurological pathologies. Post-diagnosis time can be an interesting variable in stroke, for example, that occurs suddenly, in degenerative pathologies a functional assessment is necessary because there is not always a correlation in post-diagnosis time and functional stage. In your case, postural control will depend directly on postural control. Please keep that in mind.
The authors understand your suggestion. You can be sure that we will take it into account in future research.
Once again, thank you very much for the time spent and the interest shown in this work; as well as in the positive evaluations you have given of it.
Receive a warm greeting,
The authors.